# Preparation Technology, Reactivity and Applications of Nano-Aluminum in Explosives and Propellants: A Review

**DOI:** 10.3390/nano15201564

**Published:** 2025-10-14

**Authors:** Huili Guo, Weipeng Zhang, Weiqiang Pang

**Affiliations:** Xi’an Modern Chemistry Research Institute, Xi’an 710065, China; ghlyx666666@163.com (H.G.);

**Keywords:** nano-aluminum, preparation technology, reactivity, application, explosives and propellants

## Abstract

Aluminum powder is the most commonly used metal fuel in the industry of explosives and propellants. The research progress in preparation technology, reactivity and application of nano-aluminum in explosives and propellants is systematically reviewed in this paper. The preparation technology of nano-aluminum powder includes mechanical pulverization technology (such as the ball milling method and ultrasonic ablation method, etc.), evaporation condensation technology (such as the laser induction composite heating method, high-frequency induction method, arc method, pulsed laser ablation method, resistance heating condensation method, gas-phase pyrolysis method, wire explosion pulverization method, etc.), chemical reduction technology (such as the solid-phase reduction method, solution reduction method, etc.) and the ionic liquid electrodeposition method, each of which has its own advantages. Some new preparation methods have emerged, providing important reference value for the large-scale production of high-purity, high-quality nano-aluminum powder. The reactivity differences between nano-aluminum powder and micro-aluminum powder are compared in the thesis. It is clear that the reactivity of nano-aluminum powder is much higher than that of micro-aluminum powder in terms of ignition performance, combustion performance and reaction completeness, and it has a stronger influence on the detonation performance of mixed explosives and the combustion performance of propellants. Nano-aluminum powder is highly prone to oxidation, which seriously affects its application efficiency. In addition, when aluminum powder oxidizes or burns, a surface oxide layer will be formed, which hinders the continued reaction of internal aluminum powder. In addition, nano-aluminum powder may deteriorate the preparation process of explosives or propellants. To improve these shortcomings, appropriate coating or modification treatment is required. The application of nano-aluminum powder in mixed explosives can improve many properties of mixed explosives, such as detonation velocity, detonation heat, peak value of shock wave overpressure, etc. Applying nano-aluminum powder to propellants can significantly increase the burning rate and improve the properties of combustion products. It is pointed out that the high reactivity of nano-aluminum powder makes the preparation and storage of high-purity nano-aluminum powder extremely difficult. It is recommended to increase research on the preparation and storage technology of high-purity nano-aluminum powder.

## 1. Introduction

Aluminum powder is the most commonly used metal fuel in the explosives industry, which is widely used in explosives and propellants. The application of aluminum powder in explosives can greatly improve the detonation heat and work power, as well as the damage efficiency of ammunition. Aluminum-containing explosives are widely used in ammunition for air defense weapons, ammunition for ground targets and ammunition for underwater weapons [1,2,3,4,5]. The application of aluminum powder in propellants can significantly increase their combustion heat and improve the specific impulse of rocket engines [6,7,8,9,10].

The biggest problem in the application of aluminum powder in explosives and propellants is that it is difficult for aluminum powder to react completely because of the formation of surface oxide layer, so its high combustion calorific potential cannot be fully realized. Ultrafining and nanocrystallizing micro-aluminum powder are important methods to improve the combustion completeness of aluminum powder. High-purity and high-quality nano-aluminum powder is necessary for effectively utilizing the energy characteristics of explosives and propellants. Therefore, it is very important to prepare high-purity and high-quality nanometer aluminum powder. In addition, the reactivity difference between nano-aluminum powder and micro-aluminum powder is compared in this paper.

Because nano-aluminum powder is easily oxidized, its application efficiency is seriously affected. In addition, when aluminum powder is oxidized or burned, it will form a surface oxide layer, which will prevent the internal aluminum powder from continuing to react. In addition, nano-aluminum powder will deteriorate the preparation process of explosives or propellants [11]. Appropriate coating or modification treatment is needed in order to improve these shortcomings. The latest research progress in nano-aluminum powder modification technology is also introduced in this paper, aiming to provide technical support for workers in the explosives and propellants industry.

Finally, the influence of nano-aluminum powder on the properties of explosives and propellants is also expounded.

## 2. Preparation Technology of Nano-Aluminum

### 2.1. Mechanical Pulverization Technology

#### 2.1.1. Ball Milling Method

The ball milling process utilizes the collision, impact and friction of milling balls to crush materials. Ball milling is the most commonly used mechanical pulverization technology. The shape, hardness and ratio of the milling balls, their ratio to the materials, the fill ratio of the ball milling chamber, the ball milling medium, the rotating speed and the ball milling time all affect the final product quality. The ball milling chamber can be divided into the impact area, milling area, inert area, falling area, empty throwing area and throwing area, as shown in Figure 1. Among these, the movement direction and high relative speed of the medium in the throwing area, impact area, empty throwing area and milling area will change sharply, which plays a key role in milling and crushing [12].

Yuan [13] prepared nanometer aluminum powder by the high-energy ball milling method. Anhydrous ethanol was used as the liquid medium and polyvinylpyrrolidone (PVP) as the modifier. The particle size analysis results showed that the yield of aluminum powder below 100 nm was about 22.85%. Jiang [14] prepared nano-aluminum powder by the ball milling method. Raw material aluminum powder with an average particle size of 40 microns was ground under nitrogen protection and protected by a coating agent to prevent oxidation. X-ray diffraction (XRD) test results showed that the average grain size of the prepared aluminum nanoparticles was between 25 nm and 30 nm, and the average particle size was 27.8 nm. Wang [15] prepared nano-aluminum powder with self-designed NP-V typed nano-powder production equipment. According to the X-ray diffraction (XRD) pattern, the particle size of the nanometer aluminum powder was mainly distributed in the range of 20~200 nm, the average particle size (Dn) was 98.9 nm, and the crystal size was 36.1 nm. The aluminum content, calculated according to energy spectrometry measurements, was 87.44%. Kumar [16] prepared nano-aluminum powder particles by a low-temperature milling method. It was observed that the microcrystal size of the low-temperature milling powder changed from 10 to 15 nm after 390 min of ball milling. The authors believe that the low-temperature milling process can produce high-activity aluminum powder with a particle size of about 10 nm in batches, which has important application value. Yu [17] prepared nano-aluminum powder by the ball milling method. Raw material aluminum powder with a particle size of 4~8 microns was milled in gas-phase or liquid-phase acetonitrile (ACN) under the protection of argon. The analysis results showed that about 95% of the particles were in the mass range of 50~100 nm. The authors suggested that an important factor in the effective size reduction of aluminum powder is the ability of the milling reactants to reach the crack tip quickly, and that small-molecule reactants (such as ACN) are more effective in promoting size reduction than large end-sealed molecules. Furthermore, the team in Ref. [18] also prepared nano-aluminum powder by ball milling in an ammonia and monomethylamine (MMA) atmosphere, efficiently producing nano-aluminum powder particles about 100 nm in size. Therefore, the milling medium is very important for the production of high-quality nano-aluminum powder. Salas [19] prepared nanometer aluminum powder by wet milling. The aluminum powder used was 99.8% pure, and the milling ball was an yttrium-stabilized zirconia (YSZ) ball with a diameter of 3.0 mm. Aluminum nanoparticles made from aluminum powder are produced by wet mechanical milling through a combination of different milling conditions, such as the ball–powder ratio (BPR) and the amount of solvent used. It was found that the particle size of nano-aluminum powder is about 10 nm under optimized conditions.

The analysis shows that the temperature, the material and size of the ball, the medium of the ball mill and the milling time have a significant influence on the quality of the product. Due to the different ball milling conditions adopted by different researchers, it cannot be inferred that specific ball milling conditions are optimal.

#### 2.1.2. Ultrasonic Ablation (Scrub) Method

The ultrasonic ablation (scrub) method is an emerging method for preparing some nano materials. It uses the energy-driven screw movement of an ultrasonic device to grind aluminum powder into nanoscale aluminum powder, as shown in Figure 2. Hajnorouzi [20] creatively proposed a new ultrasonic ablation (scrub) technique to prepare aluminum nanoparticles. The ultrasonic amplitude transformer bar is composed of three parts: a sandwich transducer (Langevin type), a titanium-reinforced amplitude transformer bar and a titanium screw. Ultrasonic waves are longitudinal waves that cause the titanium screw to move back and forth within a range of about 145 nm, quickly producing nanometer aluminum powder of nearly the same size. Ultrasonic ablation technology has many advantages, such as a fast process, green synthesis, no need for any other chemical precursors, homogeneous and pure products (without any solvent), stable particles, unique morphology and a narrow size distribution. Whenever the ultrasonic device is turned on, the production of nano-aluminum powder starts immediately, and all target foils can be converted into nano-aluminum powder in a matter of minutes. The particle size and preparation efficiency of aluminum powder can be changed by changing the power of the ultrasonic device and displacing the titanium amplifier and screw. No additional elements were detected in multiple analyses. Therefore, this novel method has important application value for the preparation of nano-aluminum powder.

Because ultrasonic ablation has many advantages that traditional techniques cannot match, the analysis results show that this technology has the potential to replace the existing technology and is worthy of further study.

### 2.2. Evaporation–Condensation Technology

Evaporation–condensation technology is utilized to prepare nano-aluminum powder using induction coils, plasma, arcs and other means to evaporate aluminum into gas, followed by cooling in an inert medium. Nano-scale aluminum powder can be obtained by controlling the appropriate process parameters.

#### 2.2.1. Laser Induction Composite Heating Method

Nano-scale aluminum powder was prepared by Chen [21,22] using the laser induction composite heating method. This method involves the use of high-frequency induction currents to heat and melt the metal material as a whole to reach a higher temperature, followed by laser heating evaporation and condensation. The particle size of the prepared nano-aluminum powders ranged from 15 nm to 35 nm, with an average particle size of 30 nm. The active aluminum content was measured by the X-ray diffraction K-value method [23], and the measured activity of the aluminum powder was between 40.7% and 54.5%. Duan [24] prepared nano-aluminum powder by laser induction composite heating technology, and the particle size of the prepared nano-aluminum powder was about 50 nm. The content of active aluminum reached 75.19%, based on non-aqueous solvent redox titration. Zhang [25,26] used the laser induction composite heating method to prepare carbon-coated nano-aluminum powder in situ under the protection of a methane and argon atmosphere in order to prevent oxidation of the prepared nano-aluminum powder. The prepared products had a core–shell structure, were mostly spherical, and had an average particle size of 20~40 nm.

#### 2.2.2. High-Frequency Induction Method

Wang and Guo [27,28,29] prepared nano-aluminum powder through evaporation and condensation by heating with high-frequency induction coils, followed by passivation with air to prepare nano-aluminum powder protected by a passivating layer. Transmission electron microscopy (TEM), field emission scanning electron microscopy (SEM) and X-ray diffraction (XRD) results showed that the particle sizes of the prepared nano-aluminum powder ranged from 15 nm to 60 nm, and the surface of the nano-aluminum powder was coated with alumina film 3 nm to 5 nm thick.

#### 2.2.3. Arc Method

The arc method, or arc discharge method, vaporizes aluminum rods or wires through a high-density current pulse to generate nano-scale aluminum particles. An aluminum rod or wire can be used as a moving anode or a stationary cathode. The medium can be water, alcohol, nitrogen or liquid nitrogen. The particle size and crystal type of the prepared nano-aluminum powder can be controlled [30] by adjusting the current and cooling medium. Liang [31] prepared high-purity nano-aluminum powder by the direct current arc plasma evaporation method and characterized the sample with a specific surface area analyzer and scanning electron microscope (SEM). The raw material used was a 99.91% pure aluminum block, and the working gases were argon with over 99.99% purity and 99.5% hydrogen. The preparation process was as follows: the raw metal aluminum block was placed in a crucible and vaporized by arc heating. Inert gas, driven by a high-pressure vacuum fan, blew the aluminum vapor into the collection system and passivation system for cyclone collection. After gas and fog separation, nano-scale aluminum powder was obtained. Fan [32] prepared nanometer aluminum powder by the direct current arc method. The aluminum powder was analyzed by transmission electron microscopy (TEM), and the average particle size of the aluminum powder was 88 nm. The activity of the aluminum powder was tested by the gas volumetric method, and the test result was 72.35%. Chen [33] used the direct current arc hydrogen plasma method to prepare nano-aluminum powder under a 1.0 kPa CH_4_ and 2.5 kPa Ar atmosphere. The specific surface area of the nanometer aluminum powder was measured by the BET multilayer gas adsorption method, and the calculated particle size was 373.8 nm. Zhuang [34] prepared highly active nano-aluminum powder by the DC arc plasma method. The micro-aluminum powder used was of analytical-grade purity. The aluminum powder was prepared by current arc plasma equipment under the protection of inert gas, collected and sealed with anhydrous ethanol to isolate the air and then modified by a KH-550-typed coupling agent. The prepared nano-Al powder was regular spherical or quasi-spherical, and the particle size distribution was between 20 and 100 nm.

Compared with the traditional evaporation–condensation method, the arc preparation process for nano-powder is environmentally friendly. The prepared powder products have high purity, small average particle sizes and controllable shapes and sizes, making it a nanomaterial preparation method with great application potential.

#### 2.2.4. Pulsed Laser Denudation

Pulsed laser denudation [35] is another kind of evaporation method. The high-purity aluminum target is first heated to its boiling point by a laser pulse, forming atoms containing the vapor of the plasma target. Then, the plasma expands adiabatically and cools by gas to generate nanometer-sized aluminum powder particles. Many factors affect the quality of the prepared aluminum powder, including the laser energy, laser wavelength, laser pulse width, liquid medium type and denudation time. Huang [36] used a high-frequency nanosecond pulse power supply to produce high-energy nanosecond pulses with a maximum voltage of about 3000 V to impact aluminum targets and prepare nano-scale aluminum particles. It was found that the particle size of the prepared nano-aluminum powder decreased with increases in the flow rate of the cooled inert gas. The particle size of the metal nanoparticles also decreased gradually with decreases in discharge frequency. The particle size of nanometer aluminum powder prepared under optimized conditions was about 52.9 nm. Mathe [37] prepared nano-aluminum powder by applying laser plasma impact to an aluminum block. The particle size of the aluminum powder reached 50 nm. The aluminum powder was also passivated by air and palmitic acid in order to avoid excessive oxidation of the aluminum powder.

#### 2.2.5. Resistance Heating Condensation Method

Ma [38] prepared nanometer aluminum powder by the high-efficiency vacuum resistance heating and condensation method. The raw material used was aluminum with a purity greater than 95%. The aluminum powder was placed in a high-pressure vapor deposition system, heated to a certain temperature by high-efficiency resistance in a vacuum environment. The aluminum underwent a liquefaction process, and then rapidly evaporated and vaporized. Nanometer aluminum powder with an average particle size of 80 nm was obtained after gas-phase condensation and cold deposition.

#### 2.2.6. Wire Explosive Crushing Method

The wire explosion method [39] is another physical method for preparing nanomaterials. First, the explosion chamber is pumped to a high vacuum, and then filled with high-purity inert gas to a certain pressure. The voltage of the accumulator is adjusted to keep the whole system in a stable state. An aluminum wire is fed into the explosion chamber through a wire feeding device to control its explosion frequency. The aluminum wire explodes instantaneously through plasma discharge, forming a highly dispersed aluminum powder that can reach the nanometer scale. This method still uses high temperatures generated by currents to explode an aluminum wire and prepare aluminum powder, and the process involves passing a large current through the wire, resulting in joule heating, conductor temperature rise, liquid phase heating, melting, and finally reaching the boiling point and overheating. Then, the liquid phase becomes a gas phase [40,41], so this technique can still be classified as an evaporation condensation method. Zeng [42] prepared nano-aluminum powder by the wire explosion method. The preparation system is first filled with argon gas of 99.99% purity at standard atmospheric pressure. The high-voltage discharge causes the aluminum wire to explode 120 times per minute. Then, the air is extracted and filtered for separation. The diameter of the aluminum wire used was 0.35 mm, and the purity was higher than 99.99%. Transmission electron microscopy (TEM) analysis showed that most of the final prepared aluminum powder particles had diameters of about 50 nm. Fan [32] prepared nano-aluminum powder by the electric explosion method. The prepared aluminum powder particles were nearly spherical with an average size of 81 nm, according to transmission electron microscopy (TEM) analysis. The activity of the aluminum powder was tested by the gas volumetric method, and the test result was 71.02%. Lv [43] studied the reaction kinetics of preparing nanometer aluminum powder by the electric explosion method. The results showed that deposition energy can be preferentially converted not only into the internal energy of the material but also into explosive kinetic energy. The metal wire goes through mechanical fragmentation, phase explosion, dual-mechanism explosion and supercritical explosion with increases in the deposition energy. The particle size of the explosive product decreases and the gas rate increases with increases in deposition energy. Zhao [44] prepared aluminum powder by the wire explosion method. The electron temperature of aluminum wire electroexplosive (EEW) plasma in argon was 1~2 eV, and the electron density was about 1019/cm^3^. The author mainly discussed the mechanism of the process, but did not share the specific particle size of the prepared product. Antony [45] prepared nano-aluminum powder by the wire explosion method. Wire explosions were carried out in different environments, namely argon (Ar), helium (He) and nitrogen (N_2_) at 25, 50 and 100 kPa pressures. The average diameter of the final prepared aluminum powder was in the range of 30~45 nm. Li [46] prepared nano-aluminum powder by the wire explosion method. The experimental system consisted of a 50 kA, 5 μs pulse current generator, aluminum wire electric explosion (EEW), a gas circulation system and a particle collection system. It was observed that, using different diameters of aluminum powder, almost all Al nanoparticles were basically spherical, with average particle sizes of 80.2 nm and 91.5 nm, respectively, without and with a 3.5 cm shunt gap. The average particle size of the nanoparticles decreased with increases in deposition energy. Energy transferred from the capacitor to the explosive aluminum during the partial reheating process may affect the characteristics of the nanoparticles, reducing their average particle size.

The evaporation–condensation process is relatively simple, produces easily separable products and is suitable for industrial large-scale production. However, the cost of equipment and production are relatively high compared with the ball milling method.

### 2.3. Chemical Reduction Technology

#### 2.3.1. Gas-Phase Pyrolysis Method

Kaplowitz [47] prepared nano-aluminum powder by gas-phase pyrolysis. The raw material used was TiBAl. Aluminum volatilizes during heating under an Ar atmosphere. The particle size of the prepared nano-aluminum powder can reach 87 nm, and the purity is about 64%. Although the quality of nano-aluminum powder prepared by this method is not excellent, it provides a unique idea for preparing nano-aluminum powder.

#### 2.3.2. Solid-Phase Chemical Reduction Method

Li [48] prepared nano-aluminum powder by the solid-phase chemical reduction method. The process control agent used was 1 wt% sorbitan anhydride trioleate (Span-85). Nanometer aluminum powder (n-Al) was prepared by chemically reacting lithium aluminum hydride with anhydrous aluminum chloride at a molar ratio of 3:1 in a ball mill. The results of scanning electron microscopy (SEM) and high-resolution transmission electron microscopy (TEM) showed that the prepared nano-aluminum powders were basically spherical. It can be seen from the particle size distribution diagram that the particle size of the nano-aluminum powder was mainly distributed between 45 and 65 nm, with less agglomeration. Chu [49] designed a new preparation method for nanometer aluminum powder. First, 200 mg aluminum powder (99%) with high purity was reacted with naphthalene in tetrahydrofuran (THF) solution under the protection of an inert gas to form an intermediate, and then the product was filtered and transferred to a special reaction bottle, where the intermediate was heated under vacuum and then cooled to room temperature. The particle sizes of the prepared nano-aluminum powder ranged between about 30 and 45 nm. Beaudette [50] proposed a new method for preparing nano-aluminum powder, producing 11 nm aluminum powder particles by plasma excitation of AlCl_3_ and Ar.

At present, the preparation efficiency of this method is low and needs to be optimized, but its provides a new idea for preparing nanometer aluminum powder.

#### 2.3.3. Solution Chemical Reduction Method

The solution chemistry method [51] can generate nano-scale aluminum powder by chemically reacting oxidant and reducing agent in solution. AlLiH_4_ is commonly used to react with aluminum chloride and can also be prepared by reacting with oxidants and other aluminum-containing reducing agents. Li [52] prepared a series of nano-sized aluminum powders with different particle sizes by the solution chemical reduction method. AlCl_3_ and LiAlH_4_ were used as raw materials, the solvents used were mesitylene, toluene, pyridine and n-methylpyrrolidone. Redox reactions occurred at different temperatures. In addition, different concentrations of organic amines were used to control the preparation process. The results of SEM analysis showed that the particle sizes of the aluminum powder prepared by changing the preparation conditions were between 20 and 100 nm. Li [53] prepared nano-aluminum powder by the solution chemical reduction method. Nano-aluminum powders were prepared under inert gas protection using toluene as the solvent and anhydrous AlCl_3_, N(Et)_3_ and LiAlH_4_ as the raw materials. The results of SEM and TEM analysis showed that the prepared aluminum nanoparticles were spherical and the particle size ranged from 80 nm to 120 nm. The results of gas volumetric analysis showed that the average content of active aluminum was 75.5%. Wang [54] used 1,3,5-tritylene as the solvent, anhydrous AlCl_3_ and LiAlH_4_ as the raw materials, and triphenyl phosphorus as the coating agent to effectively protect the aluminum powder from oxidation. The reaction was heated to 164 °C under magnetic stirring. TEM analysis showed that the average particle size of the prepared nano-aluminum powder was 70 nm. The particle size of the prepared aluminum powder can reach 20~30 nm when oleamine is used as the coating agent. In order to improve the quality of the prepared nano-aluminum powder, Wang also used anhydrous AlCl_3_ and LiAlH_4_ as raw materials to prepare the precursor 4AlH_3_ · Et_2_O in a mixed solution of ether and toluene, then catalyzed the decomposition to prepare nano-aluminum powder. The particle size of the prepared nano-aluminum powder was about 20~30 nm when oleamine was used as the coating agent. The particle size was also more uniform. Yu [55] catalyzed the decomposition of AlH_3_ in tetrahydrofuran solution with titanium isopropyl alcohol as the catalyst, obtaining colloidal stable monodisperse aluminum nanospheres with uniform sizes in the range of 85~200 nm. E Xiu [56] also used this method to prepare nano-aluminum powder, with particle sizes of about 16 nm under optimized conditions. McClain [57] prepared nano-aluminum powder by the solvent reduction method, easily synthesizing high-purity aluminum nanocrystals with a controllable size range from 70 to 220 nm in diameter. Size control was achieved by simply modifying the proportion of solvent in the reaction solution. Li [58] prepared nano-sized aluminum powder particles by the template method. Small aluminum nanoparticles (approximately 10 nm in diameter) were easily synthesized by catalytic decomposition of alumalane precursors using nanoscale cavities in a perfluorinated monomer membrane (Nafion-117, DuPont, Wilmington, DE, USA) as a template. This template synthesis may represent a new way to stabilize aluminum nanoparticles and related high-energy nanomaterials, providing a new idea for the preparation of a new type of template material—nanometer aluminum powder composite materials. Klein [59] proposed a thermal decomposition method starting with triisobutyl aluminum (TIBAL) as a precursor. TIBAL is refluxed at a high boiling point with diphenyl ether as the solvent and nickel, ruthenium or silver as the seed for metal nanoparticles. The resulting particle size was about 100 nm. Klein [60] also prepared nano-aluminum powders using H_3_AlNMe_2_Et or H_3_AlNEt_3_ as the precursor, alcohol salts of Ti (such as Ti(OiPr)_4_)) as the catalyst and non-polar solvents such as toluene as the reaction medium. The optimal duration for complete conversion was less than 15 min, and the optimal temperature was between 90 and 100 °C. Under optimized conditions, the particles of the prepared aluminum powder are less than 100 nm in size and basically spherical. Ghanta [61] synthesized nano-aluminum powder by the chemical method. Aluminum acetylacetonate [Al(acac)_3_] was reduced by lithium aluminum hydride (LiAlH_4_) in tritoluene at 165 °C with particle sizes between 50 and 250 nm. Cui [62] chemically synthesized nano-aluminum powders with triphenylphosphine (PPh_3_) shells. The raw materials used were AlCl_3_ and LiAlH_4_, the solvent was tritylene and the reaction temperature was 164 °C. The results of transmission electron microscopy (TEM) and high-resolution transmission electron microscopy (HRTEM) showed that the size of the prepared nano-aluminum powder ranged from 50 nm to 120 nm. Thomas [63] prepared 1,2-epoxy-9-decene terminated nano-aluminum powder by the chemical method. Nanometer aluminum powder was prepared by decomposing AlH_3_ in toluene. Transmission electron microscopy confirmed that these spherical nanostructures (25 nm in diameter) embedded in a covalently bound polymer matrix acted as a preventive barrier against water/air (H_2_O/O_2_) degradation, allowing the nano-aluminum powder to remain highly active for 6 weeks.

Some researchers have used the chemical reduction method to prepare nano-aluminum powder. The advantages of this method are that it is flexible and can be used for specific scenarios [63], such as significantly improving polymer properties. However, the process is complex, product separation is relatively difficult and the cost is high, so it is more suitable for laboratory-scale preparation and not suitable for large-scale mass production.

### 2.4. Ionic Liquid Electrodeposition Method

Ionic liquid is a room-temperature molten electrolyte, usually composed of imidazole, pyridine, other model cations and organic/inorganic anions. This special structure gives ionic liquids advantages such as a low melting point, high conductivity, a wide electrochemical window, non-volatility, non-flammability, etc. Recent studies have shown that the electrodeposition method can realize the preparation of nanoparticles in some ionic liquids. This new preparation process for nano-aluminum operates at low temperatures, has low energy consumption, provides good safety and has unique technical advantages over the traditional process. Figure 3 shows the microscopic electrodeposition process of aluminum on an electrode surface [64].

Nano-aluminum powder coating can be directly deposited onto accelerant or energetic material through a suitable process to increase the closeness of their contact, thereby increasing the reactivity of the aluminum. Abbott [65] prepared a nanometer aluminum powder coating. The effects of toluene and lithium chloride in [Bmim] Cl/AlCl_3_ ionic liquid prepared by 1-butyl-3-methylimidazole chloride ([Bmim] Cl) on the morphology of nanometer aluminum powder coating were studied. The results showed that bright nanometer aluminum powder coating can be obtained by electrodeposition after adding toluene. Li and Wang [66,67] studied the effects of potassium chloride, choline chloride, tetramethylammonium chloride and toluene on the morphology of coating in [Bmim] Cl/AlCl_3_ plasma liquid. The results showed that proper addition of choline chloride, tetramethylammonium chloride or toluene in ionic solution produces smooth and fine nanometer aluminum powder coatings. Yin and Liu [68,69] used a similar ionic system to prepare nano-aluminum powder. [Bmim] Cl/AlCl_3_ and [Amim] Cl/AlCl_3_ systems were used to successfully prepare nano-aluminum powder deposition layers by the low-temperature electrodeposition process. In addition, studies in this area [70,71,72,73,74] have used different systems to prepare deposited nano-aluminum with different specifications.

The particle liquid electrodeposition method is used to deposit nano-aluminum powder onto materials, enhancing the binding between the aluminum and the applied material. It can be inferred that more complete aluminum reaction can occur under redox reactions. This method of preparing nano-aluminum is very rare in the propellants and explosives industries. More research is still ongoing.

In conclusion, the existing preparation technologies and methods for nano-aluminum include physical and chemical methods. Table 1 presents a comparison of the advantages and disadvantages of different preparation methods for nano-aluminum powder.

From the advantages and disadvantages of different preparation methods for nano-aluminum powder in Table 1, physical methods and the ionic liquid electrodeposition method are more suitable for industrial production. The ball milling method is better in terms of process simplicity and production cost, and the ultrasonic ablation method and ionic liquid electrodeposition method are better in terms of the quality of prepared products. The appropriate preparation method can be selected according to product use and production cost.

## 3. Reactivity Differences Between Nano-Aluminum and Micro-Aluminum

The explosives and propellant industries mainly use aluminum powder with an average particle size of 5~10 μm. If the average particle size of aluminum powder is reduced to 100 nanometers, the specific surface area will increase by 2500~10,000 times, which undoubtedly has a significant impact on its reactivity. Researchers have also conducted in-depth studies on the differences in reactivity between nano-scaled aluminum powder and micrometer-scaled aluminum powder.

There are two main indexes to determine the efficiency of aluminum powder in explosives and propellants. One is related to chemical thermodynamics, that is, under what conditions aluminum powder begins to react. The other is related to chemical kinetics, the reaction rate and degree of aluminum powder under set conditions.

### 3.1. Properties of Nano-Aluminum Powder

The conditions under which aluminum powder begins to react can be investigated using ignition tests and thermal analysis, with the test results demonstrating the difference in reaction activity between nano-aluminum powder and micron-sized aluminum powder.

#### 3.1.1. Differences Between Nano-Aluminum Powder and Micron-Sized Aluminum Powder in Ignition Tests

Ignition performance determines whether the redox reaction can proceed effectively and is an important characteristic to determine whether a material is practical. Wang [75] studied the difference in ignition ability between nano-aluminum powder and micro-aluminum powder. Nano-aluminum powder with an average diameter of 80 nm and micro-aluminum powder with an average diameter of 7 μm were placed in 3 wt% polyacrylamide aqueous solution. First, the sample was placed in a quartz tube, and a certain voltage was applied to both ends of the ignition column to heat the resistance wire for ignition. The results showed that the nano-aluminum powder was ignited when the pressure was greater than 1 MPa, and the micro-aluminum powder was ignited when the pressure was greater than 2.5 MPa. This indicates that the ignition activity of nano-aluminum powder is significantly higher than that of micro-aluminum powder. Chen [76] studied the ignition characteristics of several aluminum powders. The results showed that the minimum ignition energy for 35 nm aluminum powder and 100 nm aluminum powder was less than 1 mJ, however, the minimum ignition energy for 40 μm aluminum powder was 59.7 mJ. Therefore, the ignition performance of nano-aluminum powder is far superior to that of micro-aluminum powder. In addition, the lower explosion thresholds for 35 nm, 100 nm and 40 μm aluminum powders are 40 g m^−3^, 50 g m^−3^ and 65 g m^−3^, respectively. Therefore, the smaller the particle size of aluminum powder, the lower the ignition threshold. Li [77] studied the differences in combustion characteristics between micro-aluminum powder and nano-aluminum powder. The average particle size of the micro-aluminum powder was 5 μm, and the thickness of the oxide shell was about 6.7 nm. The average particle sizes of the nano-aluminum powder were 80 nm and 120 nm, and the thickness of the oxide shells were about 1.5 nm and 1.2 nm respectively. The ignition and combustion performance of aluminum powder with different sizes was experimentally studied using a CO_2_ laser ignition device. The experiment was carried out in a power density range of 50.3~267.6 w/cm^2^. Figure 4 shows the relationship between the ignition delay time and the heat flux q, and Table 2 shows photos of the ignition and combustion process of the above samples at the same heat flux (q = 123. 3 W/cm^2^). Lx-1 to Lx-6 represent aluminum with particles of 5 μm, 80 nm, 120 nm, 5 μm (25 wt%) + 80 nm (75 wt%), 5 μm (50 wt%) + 80 nm (50 wt%) and 5 μm (75 wt%) + 80 nm (25 wt%), respectively. Figure 4 and Table 2 show that the ignition delay times of aluminum powder with different particle sizes and proportions decreased with increases in laser heat flux. At the same time, under the same heat flux, the ignition delay time of the micro-aluminum powder was much longer than that of the nano-aluminum powder. With the increase in nano-aluminum powder content, the combustion flame became brighter and brighter, and the range of spot diameter gradually increased. When only nano-aluminum powder was used, the flame was the brightest and the spot diameter of fire was the largest, indicating that the combustion was the most intense. When only micro-aluminum powder was used, the flame was the darkest and the spot diameter of fire was the smallest, indicating that the combustion was the slowest. Jin [78,79] studied the differences in ignition characteristics between micro-aluminum powder and nano-aluminum powder. Three particle sizes of micro-aluminum powder were selected, which were 2.9, 6.1 and 10.8 μm. Three particle sizes of 56.0, 74.4 and 93.4 nm were also selected for the nano-aluminum powder. The ignition test results showed that the ignition delay times for the micro-aluminum particles were 3.56 ms, 5.87 ms and 8.03 ms, respectively, and the ignition delay times increased with increases in particle size. The ignition delay times for the nano-aluminum were 2.08 ms, 1.84 ms and 1.81 ms, respectively, and the ignition delay time decreased with increases in particle size. The combustion process of nano-aluminum with different particle sizes was measured by synchronous data acquisition using a high-speed camera and infrared thermal imager, and 93.4 nm nano-aluminum powder and 10.8 μm micro-aluminum powder were selected. The results showed that it took 800 ms for the nano-aluminum to reach the highest temperature, and 2375 ms for the micro-aluminum, so the temperature rise rate of the nano-aluminum was much faster than that of micro-aluminum. Martin et al. [80,81,82,83,84] obtained the same research results through similar research methods. Similar results were found by Uhlenhake [85], where 20 wt.% of n Al and μAl particles were mixed in PVDF, and discs was manufactured by melting wire manufacturing technology. The nAl/PVDF printing disc ignited at a minimum ignition power (MIP) of 4.1 W, while the μAl/PVDF disc needed 9.5 W to ignite.

#### 3.1.2. Differences in Nano-Aluminum Powder and Micron-Sized Aluminum Powder Based on Thermal Analysis

Thermal stability represents whether the chemical properties of materials are stable in a certain environment, which is closely related to the storage performance and reaction performance of materials. The thermal behavior of aluminum powder can be characterized by differential scanning calorimetry (DSC), thermogravimetric–differential thermal analysis (TG-DTA) and gas phase analysis combined with thermogravimetric analysis (TGA); it can also be characterized by heating rate calorimetry (ARC). Wang [86] studied the influence of nano-aluminum powder and micro-aluminum powder on the thermal stability of mixed explosives. Three kinds of aluminum powders with particle sizes of 40 nm, 3 μm and 35 μm were selected, and the effective aluminum contents were 84.24%, 90.57% and 93.29%, respectively. Octagon (HMX)-based mixed explosive was prepared, and the content of aluminum powder was 35 wt%. Differential scanning calorimetry (DSC) analysis showed that the smaller the particle size of aluminum powder, the lower the initial reaction temperature and peak reaction temperature. The activation energy was calculated using the Kissinger method, and the apparent activation energies of three kinds of mixed explosives were 331.2 kJ mol^−1^, 421.7 kJ mol^−1^ and 480.9 kJ mol^−1^, respectively. Therefore, the apparent activation energy also decreased with decreases in aluminum particle size. The results showed that the smaller the particle size of aluminum powder, the easier it is to react. Lu [87] also obtained similar results. The thermogravimetric (TG) curve showed that the weight and thermal flow of nano-aluminum powder increased sharply, indicating that the reaction rate of nano-aluminum powder was much fast than micron-grade aluminum. Chang [88] studied the reactivity differences between micro-aluminum powder and nano-aluminum powder. After screening, the particle size of micro-aluminum powder was 10~20 μm, and the average particle size of nano-aluminum powder was 100 nm. The results of dust explosion experiments of nano-particles showed that they exhibited a lower minimum explosible concentration (MEC) than those of micron particle dust clouds. The MEC value for micro-aluminum powder is 510 g cm^−3^, while for nano-aluminum powder it is 190 g cm^−3^. TGA results showed that the nanoparticles of aluminum were almost completely oxidized below 1500 K. In contrast, the reaction amount of microparticles of aluminum was less than a quarter, as shown in Figure 5. Therefore, the reactivity of nano-aluminum powder is much higher than that of micro-aluminum powder. In order to understand the differences in the reaction behavior between micro- and nano-aluminum powder in the temperature range of 25~1500 °C and in the environment of O_2_ and CO_2_, Zhou [89] conducted thermal tests on 50 nm and 1.5 µm aluminum powders. The experimental results show that the reaction of 50 nm and 1.5 µm aluminum powders with O_2_ mainly occurred at 530~575 °C and 775~1500 °C, respectively. The reactions of 50 nm and 1.5 µm aluminum powders with CO_2_ mainly occurred at 445~955 °C and 590~1130 °C, respectively, in a CO_2_ environment. The results showed that nano-aluminum powder is more reactive than micro-aluminum powder and needs a lower reaction temperature.

In Figure 5, different stages represent different polymorphic phase evolution of alumina [77]. Stage I corresponds to the formation of a regular polycrystalline layer γ-Al_2_O_3_ from amorphous Al_2_O_3._ Stage II indicates that the growth of γ-Al_2_O_3_ layer is accompanied by phase transformation, and it becomes other transitional polymorphs, such as δ-Al_2_O_3_ and θ-Al_2_O_3_, but its density is very close to γ-Al_2_O_3_, and this transformation will not significantly affect the oxidation rate; At the end of this stage, stable and more dense α-alumina begins to form. At stage III, the oxide layer is completely converted to α-Al_2_O_3_.

Regarding the reactivity differences between nano-aluminum powder and micro-aluminum powder, Li [77] stated that the ignition and combustion mechanisms of nano-aluminum particles are completely different from those of micro-aluminum powder. An evaporation process occurs during the combustion process of micro-aluminum powder. In contrast, there is no gasification process during the process of ignition and temperature rise in nano-aluminum powder, but it directly reaches the oxidation stage, resulting in complete ignition. Compared with micro-aluminum powder, nano-aluminum powder requires less energy for ignition combustion, which significantly shortens the ignition delay time. The surface oxide layer hinders the evaporation of aluminum powder and its reaction with external oxygen, so the thickness of the surface oxide layer has a very significant influence on ignition. Other researchers [79,80] think that there may be a minimum value (critical point) between 93.4 nm and 2.9 μm for sub-micron-grade aluminum powder, at which the ignition delay time is minimized and the ignition performance is optimal. Bockmon [90] also obtained similar research results.

### 3.2. Difference in Reaction Completeness Between Nano-Aluminum Powder and Micro-Aluminum Powder

Whether there is a significant difference in the reaction completeness between nano-aluminum powder and micro-aluminum powder directly determines the application prospects of nano-aluminum powder, as well as its future development. Chen [91] studied the performances of composite explosives containing micro-aluminum powder and nano-aluminum powder to metal-plate acceleration. Three kinds of aluminum powders with average particle sizes of 50 nm, 5 μm and 50 μm were selected. The content of aluminum powder in mixed explosives was 20 wt%, and LiF was used for comparison. The results showed that the reactivity of the 50 nm aluminum powder was obviously better than that of the 5 μm and 50 μm aluminum powders, especially in the early stage of the reaction, no matter driving a 0.54 mm copper plate or a 1.00 mm copper plate, as shown in Figure 6. In tests driving a 1 mm copper plate, the reactivity of all three aluminum powders was higher than that of driving a 0.54 mm copper plate, which shows that the reaction of aluminum powder in the detonation of aluminum-containing explosives is not only affected by the size of the aluminum powder but also closely related to the explosive loading conditions. Small-sized aluminum powder and strong constraint conditions are more favorable for aluminum to participate in the reaction. Yan [92] studied the differences in explosion characteristics between micro-aluminum powder and nano-aluminum powder induced by shock waves. The experiment was completed in an explosion shock tube. The results showed that the ignition delay time of nano-aluminum powder was less when the shock Mach number was the same. When the shock Mach number was greater than 2.5, the ignition delay time of the nano-aluminum powder decreased sharply. XPS results showed that the oxide layer on the surface of the nano-aluminum powder product was 35 nm thick and the oxidation degree was 92%. The thickness of oxide layer on the surface of the micro-aluminum powder product was 30 nm, and the oxidation degree was 65%. Therefore, the reaction degree of the nano-aluminum powder was higher than that of the micro-aluminum powder. Zhu [93] studied the exothermic behavior of an Al/AP (ammonium perchlorate) system. Three kinds of aluminum powders with a D50 of 40 nm, 2.6 μm and 10.7 μm were used. The results of DSC analysis showed that the smaller the particle size of the aluminum powder, the lower the peak temperature of exothermic peak and the higher the heat generated by the system, indicating that nano-aluminum powder reacts more easily and completely under the same conditions. Galfetti [94] studied the differences in combustion products of propellants containing two kinds of aluminum powder. The average particle sizes of the aluminum powder were 150 nm and 30 μm, respectively, with a formula composition of Al/AP/HTPB(hydroxyl-terminated polybutadiene) = 15/68/17. The results showed that when environmental pressure of P = 0.1 MPa was applied, the proportion of unburned aluminum in the combustion products was normalized to the aluminum mass contained in the propellant sample, with (21.4 ± 8.5)% for the propellant containing 150 nm aluminum powder and (35.1 ± 5.9)% for the propellant containing 30 μm aluminum powder. When the pressure increased to 3 MPa, the two samples decreased to (17.1 ± 3.1)% and (29.9 ± 7.9)%, respectively. The results showed that the formula containing nano-aluminum had higher combustion efficiency.

### 3.3. Surface Coating and Modification of Nano-Aluminum Powder

Nano-aluminum powder has several obvious defects. Firstly, nano-aluminum powder is easily oxidized, which seriously affects its storage and application efficiency. Secondly, due to the large specific surface area, the relatively thicker surface oxide layer prevents the internal aluminum powder from continuing to react. Thirdly, nano-aluminum powder often deteriorates the preparation process of explosives or propellants. These defects can be significantly improved by using a proper coating or modification process.

According to the different bonding properties between nano-aluminum powder and auxiliary materials, coating and modification can be divided into physical processes and chemical processes. Different coating and modified materials will have different effects on various properties of nano-aluminum powder.

#### 3.3.1. Protecting the Activity of Nano-Aluminum Powder

A large number of studies have shown that coating and modification can effectively protect the activity of nano-aluminum powder and increase its application potential. Luo [95] coated nano-aluminum powder with different contents of polyethylene glycol (PEG). The results of differential scanning calorimetry (DSC) showed that the peak temperature of thermal decomposition of the coated samples was delayed to some extent compared with the uncoated samples, but the specific temperature difference in the delayed position was not specified, which indicates that PEG has a certain protective effect on the activity of nano-aluminum powder. Hao [96] used hydroxyl-terminated polybutadiene–toluene diisocyanate (HTPB-TDI) to coat nano-aluminum powder with an average particle size of 80 nm and an activity of more than 99.99%. The results of transmission electron microscopy (TEM) showed that the coating was intact, indicating potential for long-term storage of high-activity nano-aluminum powder. Zhang [97] prepared Al/PS microcapsules by coating nano-aluminum powder with polystyrene (PS) and analyzed the activity of aluminum powder in microcapsules stored for different durations. The results showed that the contents of active aluminum were 76.07%, 76.06% and 74.81%, respectively, after newly prepared, sealed and stored in an oxygen tank for 30 days under natural conditions, indicating that nano-aluminum powder coated with PS can maintain its activity for a long time. In contrast, uncoated nano-aluminum powder was stored naturally under the same conditions, and the proportion of active aluminum was only 42.3%. Zha [98] used octadecylamine (ODA) as a raw material to prepare carbon-coated nano-composite particles. After being stored in air for one month, the active aluminum content decreased from 88.5% to 87.9%, and the activity changed little, indicating that carbon coating can effectively prevent further oxidation of nano-aluminum powder. Li [99] prepared nano-aluminum powder by the redox method and coated it with paraffin wax and perfluorotetradecanoic acid, respectively. The storage test results showed that the activity of the newly prepared composite particle aluminum powder was 80.8%, and after being stored in air for two months, the active aluminum contents decreased by 1.7% and 3.8%, respectively, indicating that both paraffin wax and perfluorotetradecanoic acid can effectively maintain the activity of nano-aluminum powder. Kwon [100] used boron to coat nano-aluminum powder and stored it in air with humidity of 70% for one year. The active aluminum content of composite particles decreased from 84% to 82%, which is very low, indicating that boron had a very good coating effect. Because boron coating can increase the combustion heat of nano-aluminum powder at the same time [101,102,103,104], this material may have very good application prospects. Hammerstroem [105] coated nano-aluminum powder with epoxy resin. The results of XRD, TG and other tests showed that the active aluminum content of nano-aluminum powder prepared under optimized conditions was as high as 94%, which may have application prospects. Liu [106] used trimethylolpropane triacrylate ((CH_2_=CHCOOCH_2_)_3_CCH_2_CH_3_) (TMPTA) as monomer raw material to coat nano-aluminum powder and successfully prepared nano-A1/PTMPTA composite particles. The results showed that the dispersion, corrosion resistance and thermal stability of the composite particles were greatly improved, and the content of active aluminum powder was about 90%. After two months of testing, the activity of nano-aluminum powder was basically unchanged after coating, while the activity of naturally passivated nano-aluminum powder had decreased by 48%. Mathe [37] studied the influence of accelerated aging on palmitic acid-coated nano-aluminum powder. Even in an environment with 90% humidity and 60 °C for 10 h, the activity of the coated aluminum powder changed little, while that of the uncoated aluminum powder decreased by about 10 percentage points.

The above research results show that the oxidation rate of nano-aluminum powder after coating modification can be greatly reduced using suitable coating materials and processes. Specifically, using boron powder to coat nano-aluminum powder and PTMPTA to coat nano-aluminum powder can improve the energy of aluminum powder or have an excellent coating effect, indicating stronger application prospects.

#### 3.3.2. Improving the Reactivity of Nano-Aluminum Powder

The reactivity of nano-aluminum powder has a very important influence on its application efficiency in explosives and propellants. Zhang [107] prepared carbon-coated nano-aluminum powder by the laser induction composite heating method. The prepared core–shell structure material had a particle size range of 20–60 nm and a shell thickness of 3–8 nm. Differential thermal analysis (DTA) showed that the core–shell structure material began to emit heat at 400 °C, which is far lower than the melting point of aluminum (660 °C) and more than 100 °C earlier than alumina-coated nano-aluminum powder. Yao [108] used perfluorotetradecanoic acid to coat nano-aluminum powder with an average particle size of 50 nm and a purity of more than 98.0%, and the coating agent content was 10 wt%. Laser ignition testing was carried out and compared with uncoated aluminum powder. It was found that, compared with uncoated aluminum powder, when the laser heat flux was low, the ignition delay time of coated aluminum powder was shorter, the combustion reaction was more intense, and the flame brightness was higher. The combustion test also showed that coated aluminum powder burned more fully. Wang [109] prepared nano-aluminum powder (nAl) coated with ammonium perchlorate (AP) by the recrystallization method, and the content of the coating agent was 10–30 wt%. It was found that when the heating rate was high, AP decomposed violently, the ignition temperature of the composites with optimized formula was about 200 °C lower than that of pure nano-aluminum powder and the weight loss rate was higher than that of uncoated nano-aluminum powder. Hao [110] prepared a nano-aluminum/perfluorooctanoic acid (Fx) composite with a core–shell structure (nAl@Fx). The results of differential scanning calorimetry (DSC) showed that the instantaneous heat flux of the nAl@Fx composites was 155 W/g at 633 °C, which is much higher than that of nAl at 573 °C. The combustion test results showed that the maximum combustion flame temperature of nAl@F-x composite reached 1366 °C, which was higher than the 1198 °C recorded for the raw nAl. Wang [111] prepared nAl/PFSA nano-composite energetic materials. The average particle size of the nAl was 50 nm, with aluminum powder content of 30 wt%, and it was compared with an Al/PTFE system with the same mass ratio. The results of differential scanning calorimetry (DSC) showed that the exothermic reaction of the nAl/PFSA system was 2675 J/g, which was about three times of that of the nAl/PTFE system, and the initial reaction temperature was 370 °C, which was much lower than that of the nAl/PTFE system (463.8 °C). The combustion-supporting effect of PFSA on aluminum powder is better than PTFE. Nitrocellulose (NC) has high energy. Wang [112] prepared Al/NC composites by the electrostatic spraying method, using aluminum powder with a particle size smaller than 50 nm, containing about 70% active aluminum and 10% NC. The results of the ignition test showed that the Al/NC composite particles prepared by the electrostatic spraying method burned more violently than untreated nano-Al particles, resulting in dazzling fire, and the ignition delay time was sharply reduced from 14 ms to 0.3 ms. Luo [95] studied the influence of polyethylene glycol (PEG) and fluorocarbon surfactant TF3721 (F-C surfactant) coating on the reaction characteristics of aluminum powder. The aluminum powder used was nano-scale, and the active aluminum content was 89.70%. The original text did not specify the proportion of combustion improver. The combustion test results showed that compared with pure n-Al, PEG/n-Al composite particles and F-C/n-Al composite particles had better combustion performance, and pure n-Al could not be ignited directly. The ignition delay time of the PEG/n-Al composite particles was 0.95 s, and that of the F-C/n-Al composite particles was 1.20 s. The burning rate of the PEG/n-Al composite particles was 0.050 cm s^−1^, and that of the F-C/n-Al composite particles was 0.065 cm s^−1^. Therefore, the F-C/n-Al composite particles had a higher burning rate, and the burning phenomenon was more intense. Therefore, both PEG and F-C have a certain degree of combustion-supporting effects. Zhu [113] studied the combustion-supporting effect of KBH_4_ coating on aluminum powder. The average particle size of aluminum powder used was 50 nm, and the purity was 80.8%. It was found that when the content of KBH4 was 3%, the initial ignition temperature was 429 °C, which was 8.54% lower than that of pure aluminum powder. The highest combustion temperature was 1404 °C, which was 1.52% higher than that of pure aluminum powder. The results showed that KBH4 had a certain combustion-supporting effect on aluminum powder. Zeng [3] modified aluminum nanoparticles by in situ grafting onto high-energy glycidyl azide polymer (GAP) and prepared composites. The diameter of the aluminum powder used was about 50 nm, and its activity was 93.6%. GAP was grafted onto the surface of nano-aluminum to form an -O-(CO-NH) chemical bond. The thickness of GAP shells can be adjusted by changing the relative ratio of reactants. On this basis, (Al@GAP)/fluorine composites were prepared. The ignition test results showed that GAP could shorten the ignition time of aluminum powder, while the (Al@GAP)/fluorine composite had a shorter ignition time and more intense flame intensity. Mulamba [114] used polytetrafluoroethylene (PTFE) to coat nano-aluminum powder with an average particle size of 80 nm. The results of the combustion test showed that the burning rate of nano-aluminum powder coated with PTFE was significantly improved because A1_2_O_3_ on the surface of aluminum powder can react exothermically with tetrafluoroethylene. Cheng [115] prepared nickel-coated nano-A1 composite particles by electroless plating using a redox reaction between nickel acetate (Ni(Ac)_2_·4H_2_O) and sodium hydroxide (NaOH) in the medium containing nano-aluminum powder. Thermogravimetric analysis (TG) showed that the oxidation initiation temperature of the composite particles was 190~260 °C earlier than that of the raw material nano-aluminum powder. Qiu [116] also prepared nickel-coated nano-A1 composite particles by a similar method and obtained similar results. Kaplowitz [117] coated Fe_3_O_4_ on nano-aluminum by gas-phase inversion and formed thermite with CuO. T-Jump ignition test results showed that the ignition temperature of the thermite coated with aluminum powder was 973 K, while that of the thermite formed without aluminum powder was 1076 K, which indicates that nano-aluminum powder coated with Fe_3_O_4_ has higher reactivity. Yang [118] prepared nano-aluminum (n-Al)@ polyvinylidene fluoride (PVDF) microspheres by electrospray deposition. The average particle size of the aluminum powder was 50 nm, and the contents of PVDF were 5, 10 and 15 wt%, respectively. Combustion tests and thermogravimetric-differential scanning calorimetry analysis showed that the combustion performance of n-Al@PVD was better than that of n-Al, and the exothermic process was more intense (sharper exothermic peak). Li [119] synthesized fluorinated polydopamine (PF) and coated it with nano-aluminum powder to prepare core–shell composite materials. The combustion rate (196.4 mm s^−1^) of the optimized formula was 8.1 times and 3.6 times than that of the original n-Al (24.2 mm s^−1^) and the physical mixed sample (54.7 mm s^−1^), respectively. He [120] used an energetic metal–organic framework (EMOF) as a metal–oxide precursor and as the oxidant in conventional nano-thermite to prepare metastable nano-composite material n-Al@EMOFs. The average particle size of the aluminum powder in the material was 80 nm. EMOF is the reaction product of 5,5-bitetrazole-1,1-diol dihydrate (DHBT) and Cu(NO_3_)_2_·3H_2_O. The results showed that the ignition temperature of n-Al@EMOFs was 301.5 °C, which is much lower than the 578 °C recorded for nano-aluminum. This indicates that this structure has a good combustion-supporting effect on aluminum powder.

Ruesch [121] studied three kinds of composite propellants of the AP/Al/HTPB system, with different aluminum particle sizes (31 μm, 4.5 μm and 80 nm, respectively), and propellant without aluminum for comparison, burning in air at 1 atm. It was found that the propellant with the smallest particle (nano-aluminum) had the highest average temperature, and the temperature changed much less with changes in the measurement position. This indicates that the nano-aluminum not only reacted more easily than the micro-aluminum, but also the propellant system containing nano-aluminum burned more stably.

The large number of studies mentioned above clearly show that proper coating and modification materials can significantly increase the chemical reactivity of nano-aluminum powder and greatly improve the application possibilities of nano-aluminum powder in explosives.

#### 3.3.3. Increasing the Reaction Degree of Nano-Aluminum Powder

The reaction extent of aluminum powder is the most critical indicator of its performance in explosives. It is evident that the higher the reaction extent, the more aluminum powder can leverage its energy advantage. Yan [122] used fluororubber F2602 with fluorine content of 66% to coat nanoscale aluminum powder, with the mass fractions of 5, 10, and 15%. fluororubber. Differential scanning calorimetry (DSC) studies found that the heat release rate of the uncoated sample was 6451 J/g, while the heat release rates of the three coated samples were 6902, 7892 and 9259 J/g, representing increases of 6.99, 22.34 and 43.53% over the uncoated sample, respectively. The results showed that as the content of fluororubber increased, the total heat release also increased significantly. Lv [123] prepared Al/PTFE core–shell-structured composite materials using chemical vapor deposition technology, with an aluminum powder particle size of 150–200 nm and an effective aluminum content of 85.3%. The results showed that the combustion heat of the Al/PTFE composite material increased by 4.1% compared to the raw nano-scale aluminum powder when tested using the oxygen bomb method. Sun [124] prepared an Al@PVDF core–shell-structured composite by electrostatic spraying with an average particle size of 69 nm, and compared it to samples prepared by mechanical mixing. The results showed that fluorinated materials improved the combustion performance of nano-aluminum powders. The PVDF-coated nano-aluminum composite particles prepared by electrostatic spraying had a higher heat release rate and more complete combustion compared to the composite prepared by mechanical mixing. Chang [125] prepared aluminum/fluorinated acrylic ester (Al/PFDMA) composite particles and aluminum/polyvinylidene fluoride (Al/PVDF) composite particles. TG/DSC analysis showed that when the polymer content was 20 wt%, the reaction heats of Al/PVDF and Al/PFDMA composite particles were 11,908 J/g and 8203 J/g, which were 68% and 15% higher than that of the raw nano-aluminum (7066 J/g). The burning rates of propellants made from these two materials were also significantly increased. Kim et al. [126,127] coated nano-aluminum powder with polytetrafluoroethylene (PTFE), which was used as both a protective layer and an oxidation promoter. The results showed that, compared with pure aluminum, the weight calorific value of the composite increased from 0.88 kJ/g to 4.80 kJ/g, more than four times. The team also prepared an Al/PVDF composite, and the exothermic enthalpy was about twice that of uncoated Al particles. Wang [128] proposed an effective surface-activation strategy for nano-aluminum powder to improve the combustion performance and energy output of nano-aluminum-based energetic materials. By coating the aluminum powder with perfluododecanoic acid, a porous AlF_3_ shell was formed on the surface of the nano-Al particles. The energy output and combustion reaction rate of the C_11_F_23_COOH-coated PTFE/nano-aluminum were 6304 J/g and 670 m/s, 3.0 and 2.6 times than that of PTFE/uncoated-nano-aluminum. Campbell [129] coated aluminum powder with perfluorotetradecanoic acid (PFTD) to prepare Al/PFTD composites with different proportions, with an average particle size 1.6 μm. Ignition tests showed that when the PFTD content was 9 wt%, the maximum temperature could reach 1826 °C, far higher than the maximum temperature of pure aluminum powder at 1310 °C. Jiang’s team [130,131] modified graphene by adopting perfluorooctyltriethoxysil-functionalized graphene oxide (CFGO) and selected aluminum powder with a nominal particle size of 70 nm and an active aluminum content of 72.5 wt% to prepare nAl/CFGO (80/20 wt%) composites. Constant-volume ignition tests showed that nAl/CFGO exhibited the most intense combustion and the highest combustion temperature, with peak pressure and a pressure rise rate more than seven times and six times higher than those of the original nano-aluminum powder, respectively. In addition, the research team modified nanoscale Al with perfluoroalkyl silanes, and similar research results were obtained. Wang [132] prepared aluminum powder composites coated with fluororubber F2311. The aluminum powder was prepared by the electroexplosive method with an average particle size of about 200 nm. The results showed that compared with the raw nano-Al material, the apparent activation energy of Al/F2311 was reduced by 4 kJ/mol, and the temperature of the first exothermic peak was advanced by about 10 °C. Furthermore, the heat release was nearly twice that of the raw nano-aluminum material. Moreover, the research results of Huang [133] also showed that when using fluororubber F2314 as a binder to prepare n-Al/CL-20 microspheres, F2314 promoted the oxidation reaction of aluminum powder, resulting in higher heat release. Uhlenhake [134] prepared Al/THV composites (THV is a terpoly of tetrafluoroethylene, hexafluoropropylene and vinylidene fluoride), with an average particle size of 80 nm and a of 70 wt% Al content. The Al/THV composites were then incorporated into the AP/HTPB propellant system. When the composite material accounted for 5 wt%, the burning rate was 2.1 times than that of the original propellant. When the composite material accounted for 15 wt%, the burning rate was 4.7 times than that of the original propellant. Additionally, the team also prepared Al/PVDF films, which showed similar effects [135]. Tang [136] used fluorinated polyurethane (FPU) as a binder to prepare energetic composites containing aluminum nanoparticles (with an average particle of 100 nm). The explosion calorimetry test showed that when the fluorine content in the composite was only 3.70 wt%, the explosion calorimetry of the composite increased from 1685 kJ kg^−1^ for the non-fluorinated system to 3222 J kg^−1^, an increase of 91.2%. The authors calculated that if the additional heat of explosion was solely due to the formation of AlF_3_, it would only be 482 kJ kg^−1^. Therefore, the fluorine in the fluorinated materials not only undergoes oxidation reactions but also acts as a catalyst. When the fluorine contents in the composite were increased to 4.55, 5.24, 3.2 and 6.79%, the explosion calorimetry rates were 2973, 2926, 3032 and 3037 kJ kg^−1^, respectively, and did not continue to increase with the fluorine content. It can be inferred that fluorine acts as a catalyst.

A large number of studies have shown that suitable coating and modification materials have a great influence on the reaction degree of nano-aluminum powder, and suitable materials can even increase the heat release of raw aluminum powder by several times.

The combustion mechanism of aluminum powder is significant for its practical application. Chen and his team [137] explored the combustion mechanism of single micron-sized aluminum particles using a numerical model. The research results showed that the diffusion coefficient of oxidizers, the activation energy of surface kinetics and the evaporation coefficient (α) of aluminum impact the particle temperature the most. The burning time is most sensitive to the activation energy of surface kinetics. For some important issues, such as the combustion mechanism, combustion thermodynamics and combustion kinetics of nano-aluminum particles, the particle sizes and effective aluminum contents of aluminum powder used by different researchers have varied significantly. Additionally, most researchers have not shared the effective aluminum content used, making it difficult to obtain convincing views. Therefore, this article will not discuss this topic in detail.

## 4. Differences in Energy Release Characteristics Between Nano-Aluminum Powder and Micro-Aluminum Powder in Explosives and Propellants

### 4.1. Differences in Energy Release Characteristics Between Nano-Aluminum Powder and Micro-Aluminum Powder in Explosives

Many researchers have studied the differences in nanometer aluminum powder and micro-aluminum in mixed explosives. Liu [138] studied the reactivity differences between micro-aluminum powder and nano-aluminum powder in mixed explosives by the flyer push test. The results showed that the initial reaction time of small particles was earlier than that of large particles. Cao [139] studied the influence of nano-aluminum powder on the detonation velocity of RDX-based explosives. The results showed that the detonation velocity of samples containing nano-aluminum powder was higher than those of containing micron-sized aluminum powder when the content of aluminum powder was below 10%. Liu [140] studied the influence of nano-aluminum powder on explosive detonation pressure. The average particle size of the aluminum powder was 170 nm, the main explosive was RDX and 5 wt% wax was used as the binder and desensitizer. The results showed that when the content of nano-aluminum powder was 5 wt% and 10 wt%, the explosion pressure was higher than that of the composite without aluminum powder. In addition, Brousseau [141] studied three percentages of aluminum, namely 0%, 20% and 30%. The results showed that the explosive charge containing nano aluminum powder had higher detonation velocity than that containing micro-aluminum particles. With increases in the charge diameter, the detonation velocity difference decreased. Bai [142] carried out flyer experiment driven by the explosion of mixed explosives containing aluminum powder with different particle sizes. Three kinds of aluminum powders with particle sizes of 50 nm, 5 μm and 50 μm were selected. The results showed that the free surface velocity of the flyer was higher for aluminized explosives containing aluminum powder with a particle size of 50 nm than that of aluminized explosives containing aluminum powder with particle sizes of 5 μm and 50 μm when driving a copper plate. Wang [143] studied the influence of nano-aluminum on the metal driving ability of RDX-based mixed explosives by cylinder tests. The total amount of aluminum powder in the composite was 20 wt%. The particle size of the nano-aluminum powder was 100 nm~200 nm, which was used to replace the micro-aluminum powder in the composite. The results showed that the expansion force of detonation products increased by 10.2% and 5.5%, respectively, after the aluminum powder in the composite was replaced by 5 wt% and 10 wt% nano-aluminum powder. The author stated that the addition of nano-aluminum powder may be beneficial to adjust the rupture time of warhead shells and improve the energy utilization efficiency of charge. Huang [144] prepared explosive composites containing nano-aluminum but did not specify the particle size or effective aluminum content of nano-aluminum. The composition was HMX/micro-aluminum/nano-aluminum/binder = 57/28/5/10, and it was compared with the composite without nano-aluminum (HMX/micro-aluminum/binder = 57/33/10). The contrast formula only ignited cotton yarn 3 m away from the explosion center, while the formula containing nano-aluminum ignited the cotton yarn 5~7 m away, and the arson effect of the composite containing nano-aluminum powder was significantly improved. Elert [145] replaced micron aluminum powder with nano-aluminum powder in an explosive composite (the specific composition was not announced for confidentiality reasons), and the plate dent test showed that the dent volume on the witness plate was about twice that of the original composite. Huang [146] studied the influence of nano-aluminum on the shock wave overpressure in an air explosion test of mixed explosives. The average particle size of the nano-aluminum was 91.7 nm, and flammable fine aluminum powder was used for comparison. The test results showed that the peak value of shock wave overpressure of the sample containing nano-aluminum was much higher than that of the sample containing micro-aluminum for all the test data at all distances. This indicates that nano-aluminum has obvious advantages. Simić [147] studied the influence of nano-aluminum powder on the overpressure and impulse of air explosion shock waves. The explosive formula was HMX/AP/Al/HTPB = 45/15/20/20. Two kinds of aluminum powders with average particle sizes of 5 μm and 70 nm were selected. One composite was only micro-aluminum powder, while the other was 50% nano-aluminum powder. The air explosion test results showed that the composite with micro-aluminum powder had a lower overpressure peak in the far-field shock waves than that of the mixed aluminum powder composite. For impulse, the composite with mixed aluminum powder was higher than the micro-aluminum powder composite at all measuring points. The above experiments showed that nano-aluminum powder had a better supporting effect on shock waves. Fang [148] studied the influence of nano-aluminum powder on the explosive power of fuel–air explosives. The micro-aluminum powder used was flaky aluminum powder with a diameter of about 18 μm, a thickness of less than 1 μm and an effective aluminum content of 85%. The d_50_ of nano-aluminum powder was 100 nm, and the effective aluminum content was 88%. The results are shown in Table 3. The peak value and the maximum rising rate of explosion pressure were 0.82 MPa and 1.75 MPa s^−1^, respectively when pure micro-aluminum powder was used for the test. When adding 5% and 10% nano-aluminum powder, the peak explosion pressure increased to 1.02 MPa and 1.30 MPa, respectively, with increases of 24.4% and 58.5%. Additionally, the maximum pressure rise rates increased to 3.16 MPa s^−1^ and 3.56 MPa s^−1^, with increases of 80.6% and 103.4%, indicating the addition of nano-aluminum powder had a significant effect. Under the same ignition energy and the same environmental conditions, the maximum overpressure of fuel–air explosives showed an obvious increasing trend with increases in aluminum powder content. Nano-aluminum powder can replace some micron-sized aluminum powder, which can increase the peak explosion pressure and the rising rate of explosion pressure, improving the power of mixed explosives. Feng [149] studied the effects of nano-aluminum powder on the underwater explosion energy of RDX-based mixed explosives. The particle size of the micro-aluminum powder was 4~5 μm, and the particle size of the nano-aluminum powder was 60~80 nm. The results showed that when the total contents of aluminum powder were 30% and 35%, the total energy of underwater explosion of explosives was higher than that of explosives containing only micro-aluminum powder when mixed with nano-aluminum powder. Niu [150,151] studied the effect of nano-aluminum on the underwater explosion energy of RDX-based explosives. The average particle size of micro-aluminum powder was about 5 μm, and the average particle size of nano-aluminum powder was about 150 nm. It was found that when the total mass fraction of aluminum powder was 30%, if the mass ratio of micro-aluminum to nano-aluminum was 50:50, the total energy of underwater explosion was higher than that of the composite containing micro-aluminum alone. Therefore, nano-aluminum powder changes the energy output structure of underwater explosions of mixed explosives. Liu [152,153] studied the degree of participation of nano-aluminum powder and micro-aluminum powder in C-J surface reactions with a CL-20/Al formula system. Three kinds of aluminum powder, 200 nm, 2~3 μm and 16~18 μm in size, were used. The ratio of the reaction product particle velocity to the C-J plane particle velocity at 1 μs was tested. The results showed that the degree of 200 nm aluminum powder in the system participating in the C-J surface reaction was significantly higher than that in the other two systems, and the reaction start time of aluminum powder with a smaller particle size was earlier than that of aluminum powder with a larger particle size.

### 4.2. Differences in Energy Release Characteristics Between Nano-Aluminum Powder and Micro-Aluminum Powder in Propellants

Many studies have compared the differences between nano-aluminum powder and micro-aluminum powder regarding their propellant properties, including ignition performance, combustion performance, and particle size effects.

Jiang [154] studied the influence of nano-aluminum powder on the ignition performance of propellants. The average particle size of the nano-aluminum powder was 83 nm, and the average particle size of the ordinary aluminum was 13 μm. The research showed that the ignition threshold of the propellant containing nano-aluminum powder was several orders of magnitude smaller than that of the propellant containing ordinary micro-aluminum powder, and adding nano-aluminum powder significantly shortened the ignition delay time of the propellant. Hao [155] also showed that the smaller the particle size of aluminum powder, the shorter the ignition delay time of AP/HTPB composite solid propellants containing nano-aluminum powder. Additionally, Meda [156] prepared propellants with aluminum powders of different particle sizes. The research showed that the smaller the particle size of the aluminum powder, the lower the ignition temperature of the propellant and the shorter the ignition delay time.

Wang [157,158] studied the influence of nano-aluminum powder on the combustion performance of propellants and adopted four kinds of aluminum powder with average particle sizes of 50 nm and 100 nm, respectively: long ellipsoid, with average sizes of 53.35 nm and 105.11 nm; and flat ellipsoid, with average sizes of 55.56 nm and 110.88 nm, respectively. The composite consisted of 80–85 wt% aluminum powder, 12~17 wt% oxidant, 2 wt% binder and 1 wt% catalyst. The density of the prepared nano-aluminum powder column was about 1.8–2.0 g cm^−3^, and the column size was φ20 mm × 150 mm. It was found that the burning time of the propellant column containing 50 nm aluminum powder was obviously shorter than that of the propellant column containing 100 nm aluminum powder. When the aspect ratio was 10, the maximum burning rate of the flat ellipsoid particles was only 1.3 × 10^−13^ kg s^−1^, while that of the long ellipsoid particles was as high as 3.0 × 10^−13^ kg s^−1^, which is about 2.3 times that of the flat ellipsoid particles. Li [159] replaced 3 wt% micro-aluminum powder with nano-aluminum powder to prepare a propellant. It was found that the static burning rate of the sample increased by 22.0% at 7 MPa and 8.1% at 15 MPa. However, the dynamic burning rate increased by 14.3% when the engine was tested at 7 MPa, and increased by 9.8% when it was tested at 15 MPa. The static burning rate pressure index of the sample was 0.11 (7–15 MPa), which is 31.0% lower than that of the original sample. The dynamic burning rate pressure index obtained in the engine test was 0.22, which was 12.0% lower than that of the original sample. Ramakrishnan [160] studied the influence of aluminum powder with different particle sizes on the combustion performance of propellants. The average particle sizes of the aluminum powder were 18 μm, 250 nm, and 100 nm, respectively, and they were tested in the range of 0.2–3.1 MPa. The research showed that the combustion rates of propellants containing nano-aluminum were 2–5 times than that of propellants containing micro-aluminum. Zhang and others [161,162,163,164,165,166] applied nano-aluminum powder to propellants, which increased the burning rate of the propellants and reduced the pressure index. Galfetti [167] studied the AP/Al/HTPB solid rocket propellant system, and found that the aluminized composite solid rocket propellant containing nano-aluminum particles had a faster combustion rate than the control propellant containing the same mass percentage of micro-aluminum particles. When 20%, 50% and 100% nano-aluminum powder replaced micro-aluminum powder, the combustion rate increased by about 40%, 60% and 100%, respectively. This increase was mainly determined by the strong energy released by the oxidation of ultrafine particles near the combustion surface. In addition, the analysis of condensed combustion products showed that when the propellant formula contained nanoparticles, the combustion process was more efficient, and nano-aluminum particles reduced the unburned metal content in the condensate and increased the alumina content. This indicates that the reaction of nano-aluminum powder is more complete. Watson [168] obtained similar results. For an Al/MoO_3_ composite system, the peak burning rate of Al/MoO_3_ composite containing 50 nm Al particles was 960 m/s, while that of the same system containing 1–3 µm Al was only 244 m/s. James [169] studied the influence of nano-aluminum powder on propellant retrogression rate, and the particle sizes of the aluminum powder used were 100 nm and 20–30 μm, respectively. The composition of the system was Al/AP/HTPB = 10/70/20. It was found that the retrogression rate of the system containing nano-aluminum powder was higher than that of the propellant system containing micro-aluminum powder. Jiang [154] observed the agglomeration of combustion products of propellants containing nano-aluminum powder. After the combustion test, it was observed that the residue of propellants containing nano-aluminum powder was small and white, or mixed with a little gray, and the residue of propellants containing micro-aluminum powder was large and dark, indicating that nano-aluminum powder could burn more fully than micro-aluminum powder. Galfetti [94] found that the average size of combustion products of propellants containing 150 nm aluminum powder was 6.6 μm, while that of propellants containing 30 μm aluminum powder was 13 μm. Liu [170,171] used micro-aluminum powder with different particle sizes and obtained similar results. The particle size of agglomerates decreased with increases in the aluminum powder particle size.

Peuker [172] applied aluminum powder with different particle sizes to explosives and obtained the following conclusions: (1) Aluminum particles with particle sizes ranging from 3 to 40 μm enhance primary blasting, and the blasting driving effect does not strongly depend on particle size. (2) The quasi-static pressure measurement showed that the oxygen in the air was enough to completely oxidize aluminum powder within 10 μm. (3) In the absence of external oxygen, aluminum is usually only oxidized to 50%, but the aluminum powder with the smallest particle size (3 μm) was almost completely oxidized. These research results indicate that different application scenarios and different systems of propellants and mixed explosives may be brought into full play by using aluminum powder with different particle sizes under different environmental conditions.

The influences of nano-aluminum particles on the properties of explosives and propellants are summarized in Table 4.

Compared with micro-aluminum powder, nano-aluminum powder has higher reaction activity and a more complete reaction, so nano-aluminum powder has good application prospects. However, the high reactivity of nano-aluminum powder makes the preparation and storage of high-purity nano-aluminum powder extremely difficult; therefore, further research on preparation and storage technologies is extremely important to ensure that this material can play a greater role in many fields as soon as possible.

## 5. Conclusions

In summary, there are many ways to prepare nano-aluminum, such as mechanical crushing methods (including the ball milling method and ultrasonic ablation method, etc.), evaporation–condensation methods (including the laser induction composite heating method, high frequency induction method, arc method, pulse laser ablation method, resistance heating condensation method, gas phase pyrolysis method, wire explosion crushing method, etc.), chemical reduction methods (including the solid phase reduction method, solution reduction method, etc.) and ionic liquid electrodeposition methods. Each of these methods has its advantages and disadvantages, but many researchers have only provided the particle size of the nano-aluminum powder prepared by their team, and not the effective aluminum content. In fact, the effective aluminum content has a very important effect on the application of mixed explosives and propellants.

Some innovative nano-aluminum preparation methods, such as low-temperature ball milling, the ultrasonic ablation (scrub) method to prepare nano-aluminum powder and the template method to prepare nano-aluminum powder and active organic composite materials, may have very promising application value, and further research has been suggested.

The reaction activity and completeness of aluminum powders with different particle sizes are significantly different. A large number of studies from the perspective of ignition performance and combustion performance show that the reaction activity of nano-aluminum powder is much higher than that of micro-aluminum powder. Many studies on reaction completeness show that nano-aluminum powder achieves significantly higher reaction completeness than micro-aluminum powder.

As far as the characteristics of nano-aluminum powder are concerned, coating or modifying nano-aluminum powder can provide many advantages, such as protecting the activity of nano-aluminum powder, improving the reactivity of nano-aluminum powder and increasing the reaction heat of nano-aluminum powder.

The application of nano-aluminum powder in mixed explosives can improve many properties of mixed explosives, such as the detonation velocity, detonation heat and the peak overpressure of shock waves. The application of nano-aluminum powder to propellants can significantly improve the burning rate and the agglomeration of combustion products.

## Figures and Tables

**Figure 1 nanomaterials-15-01564-f001:**
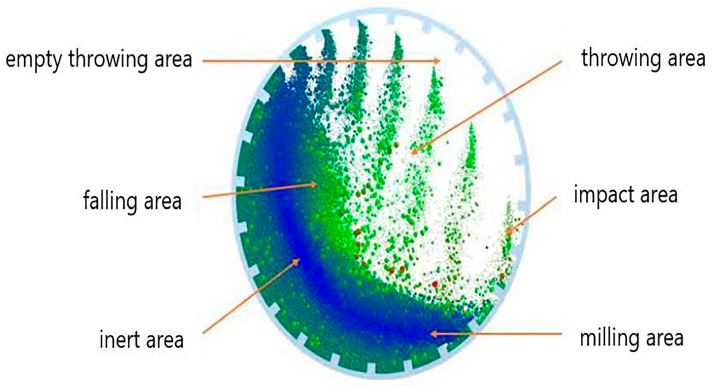
Division of movement area in ball mill chamber. Division of movement area in ball mill chamber [12]. Reproduced with permission from *Nonferrous Metals (Mineral Processing Section)* Editorial Office, 2024.

**Figure 2 nanomaterials-15-01564-f002:**
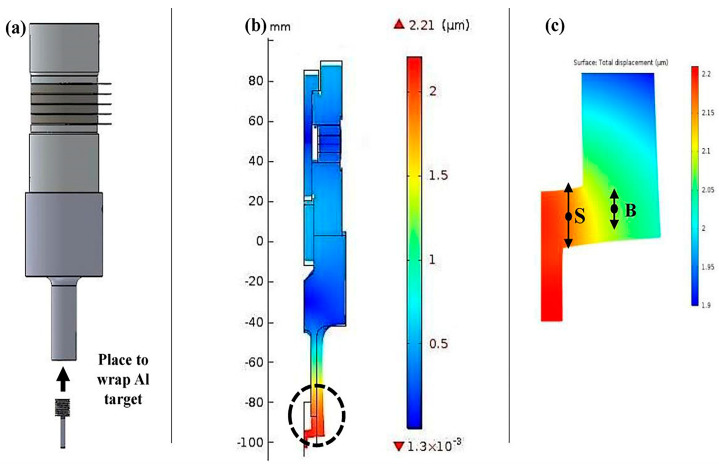
Transducer simulation. (**a**) Schematic of the transducer; (**b**) FEM analyses of the horn; (**c**) magnified dashed circle in (**b**). The arrows show the screw displacement (S) and the booster displacement (B) [20]. Reproduced with permission from Elsevier, 2019.

**Figure 3 nanomaterials-15-01564-f003:**
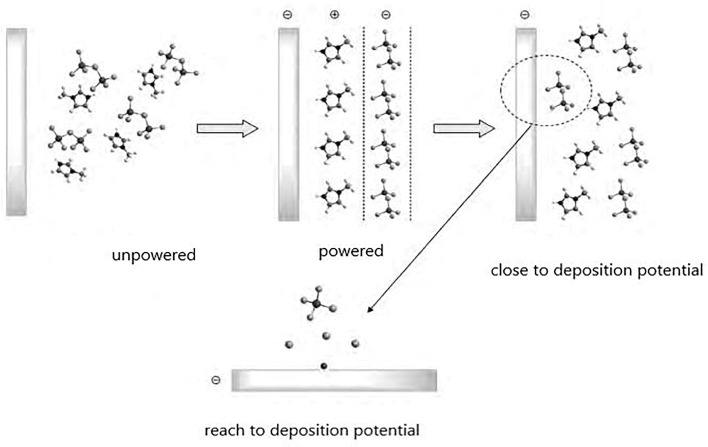
Electrodeposition process of aluminum on electrode surface [64]. Reproduced with permission from *Nonferrous Metals (Extractive Metallurgy)*, Editorial Office, 2017.

**Figure 4 nanomaterials-15-01564-f004:**
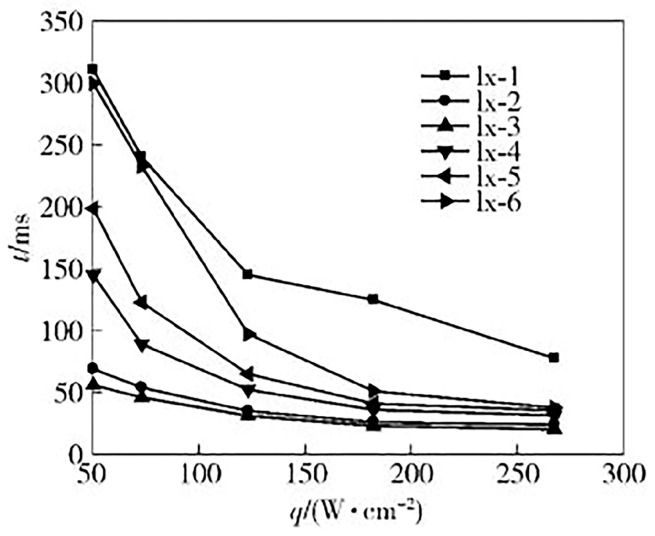
Ignition delay times for aluminum powders with different sizes and proportions as a function of laser heat flux density [77]. Reproduced with permission from *Acta Armamentarii*, Editorial Office, 2014.

**Figure 5 nanomaterials-15-01564-f005:**
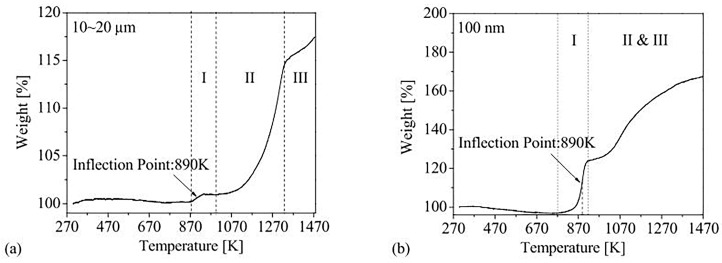
TGA results: (**a**) 10–20 μm Al particles; (**b**) 100 nm Al particles [88]. Reproduced with permission from Elsevier, 2020.

**Figure 6 nanomaterials-15-01564-f006:**
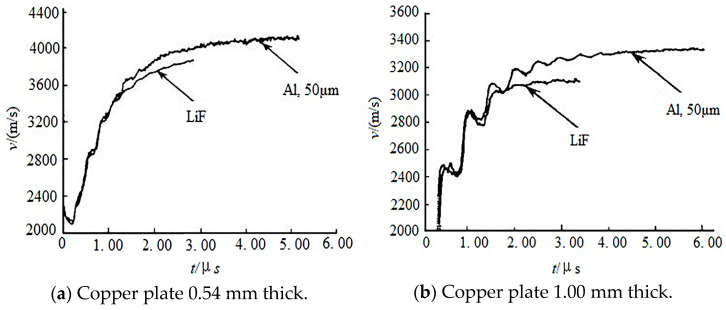
Free surface velocity of two kinds of copper plates [90]. Reproduced with permission from *Explosion and Shock Waves,* Editorial Office, 1999.

**Table 1 nanomaterials-15-01564-t001:** Advantages and disadvantages of different preparation methods for nano-aluminum powder.

Preparation Methods			Product Average Granularity	Product Purity	Manufacturability	Equipment Cost	Feasibility of Industrial Production	Environmental Friendliness
physical methods	mechanical pulverization method	ball milling method	low to medium to excellent	low to medium to good	easy	cheap	feasible	environmentally friendly
		ultrasonic ablation (scrub) method	excellent	good	difficult	high costs	needs to be researched deeply	environmentally friendly
	evaporation–condensation method	laser induction compound heating method	medium to excellent	low to medium	medium to difficult	high costs	feasible	environmentally friendly
		high-frequency induction method	medium to excellent	low to medium	medium to difficult	high costs	feasible	environmentally friendly
		arc Method	low to medium to excellent	low to medium to good	medium to difficult	high costs	feasible	environmentally friendly
		pulsed laser denudation	good	medium to good	medium to difficult	high costs	feasible	environmentally friendly
		resistance heating condensation method	good	medium to good	medium to difficult	medium to high costs	feasible	environmentally friendly
		wire explosive crushing method	low to medium	low to medium	low to medium	cheap to medium	feasible	environmentally friendly
chemical methods	gas-phase pyrolysis method		medium to good	low to medium	medium	cheap	suitable for the laboratory scale	waste
	solid-phase chemical reduction method		medium to good	low to medium	medium	cheap	suitable for the laboratory scale	waste
	solution chemical reduction method		medium to excellent	medium	medium	cheap	suitable for the laboratory scale	waste liquid
	ionic liquid electrodeposition method		nanofilms	good	medium	medium	feasible	waste liquid

**Table 2 nanomaterials-15-01564-t002:** Ignition and combustion processes for aluminum powders with different sizes and proportions (q = 123.3 W/cm^2^) [77]. Reproduced with permission from *Acta Armamentarii*, Editorial Office, 2014.

Samples	The Photos of Ignition and Combustion Processes
lx-1	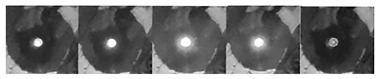
lx-2	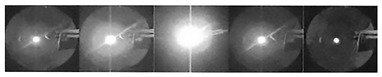
lx-3	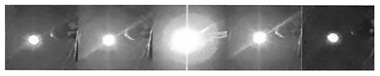
lx-4	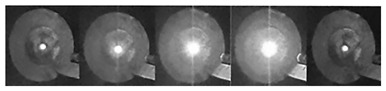
lx-5	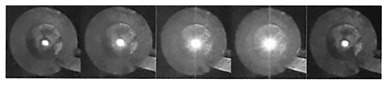
lx-6	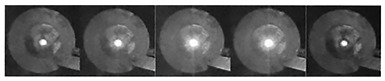

**Table 3 nanomaterials-15-01564-t003:** Explosion parameters of micro/nano-aluminum mixture [149].

No.	Concentration of Aluminum Dust/g·m^−3^	Nano-Aluminum Content/%	Maximum Pressure/MPa	(d*ρ*/d*t*)_max_/MPa·s^−1^
1^#^	344	0	0.82	1.75
2^#^	342	5	1.02	3.16
3^#^	342	10	1.30	3.56
4^#^	343	15	1.16	3.12

**Table 4 nanomaterials-15-01564-t004:** Influences of nano-aluminum particles on the energy release characteristics of explosives and propellants.

Applications	Test Method	Results	References
	flyer push test	The initial reaction time of small particles was earlier than that of large particles.	[138]
	detonation velocity	The detonation velocity of samples containing nano-aluminum powder was higher than those of containing micron-sized aluminum powder when the content of aluminum powder was below 10%.	[139]
	detonation pressure	When the content of nano-aluminum powder was 5 wt% and 10 wt%, the explosion pressure was higher than that of the composite without aluminum powder.	[140]
	flyer experiment	The free surface velocity of flyer was higher for aluminized explosives containing aluminum powder with a particle size of 50 nm than that of aluminized explosives containing aluminum powder with particle sizes of 5 μm and 50 μm.	[141]
	cylinder test	The expansion force of detonation products increased by 10.2% and 5.5%, respectively, after the aluminum powder in the composite was replaced by 5 wt% and 10 wt% nano-aluminum powder.	[142]
explosives	arson effect	The arson effect of the composite containing nano-aluminum powder was better than composite containing micro-aluminum powder.	[144]
	plate dent test	The dent volume of the composite with nano-aluminum on the witness plate was about twice that of the original composite.	[145]
	air explosion test	The peak value of shock wave overpressure of the sample containing nano-aluminum was much higher than that of the sample containing micro-aluminum for all the test data at all distances.	[146]
		The impulse of the composite with nano-aluminum powder was higher than that of micro-aluminum powder at all measuring points.	[147]
	explosion chamber experiment	The peak value and the maximum rising rate of explosion pressure of composites with nano-aluminum were better than those of composites without nano-aluminum.	[148]
	underwater tests	The total energy of underwater explosion was higher than that of the composite containing micro-aluminum alone.	[149,150,151]
	C-J surface reaction tests	The degree of aluminum powder in the system containing nano-aluminum powder participating in C-J surface reaction was significantly higher than that in systems containing micro-aluminum powder, and the reaction start time of aluminum powder with a small particle size was earlier than that of aluminum powder with a large particle size.	[152,153]
propellants	ignition tests	The ignition threshold of propellant containing nano-aluminum powder was several orders of magnitude smaller than that of propellant containing ordinary micro-aluminum powder, and adding nano-aluminum powder can significantly shorten the ignition delay time of propellant. The smaller the particle size of aluminum powder, the shorter the ignition delay time, and the lower the ignition temperature of the propellant.	[154,155,156]
		The combustion rate and retrogression rate of propellant containing nano-aluminum were much faster than those of propellant containing micro-aluminum.	[157,158,159,160,161,162,163,164,165,166,167,168,169]
	combustion rate experiment	The residue of propellants containing nano-aluminum powder was small and white, or mixed with a little gray, and the residue of propellants containing micro-aluminum powder was large and dark, indicating that nano-aluminum powder could burn more fully than micro-aluminum powder.	[95,154,170,171]

## Data Availability

Not applicable.

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
