# Peer review of "Preparation Technology, Reactivity and Applications of Nano-Aluminum in Explosives and Propellants: A Review"

_nanomaterials, 2025, doi:10.3390/nano15201564_

Round 1

Reviewer 1 Report

Comments and Suggestions for Authors

The article is very interesting and you can see that a lot of work has been put into it. However, due to the fact that it is a review article, the collected knowledge should include the results of research conducted all over the world. Meanwhile, the vast majority of cited items (almost all) are by authors from one region of the world. Do scientists in other countries not deal with this topic? This is a serious deficiency that needs to be corrected.

Author Response

Response to Reviewer 1​​

​​Comment 1:​ The article is very interesting and you can see that a lot of work has been put into it. However, due to the fact that it is a review article, the collected knowledge should include the results of research conducted all over the world. Meanwhile, the vast majority of cited items (almost all) are by authors from one region of the world. Do scientists in other countries not deal with this topic? This is a serious deficiency that needs to be corrected.

​​Response:​ Thank you for this important observation. In fact, the preparation technology of nano-aluminum powder and its application in explosives and propellants are a relatively frontier research direction all over the world. It took us more than half a year to search all the important databases, including but not limited to SCI, EI, Elsevier, Wiley and CNKI, and read all the manuscripts carefully to sort out this review. China's manuscripts account for a large proportion, but this is the case. We didn't deliberately choose the work of China's author. Please understand.​​

Reviewer 2 Report

Comments and Suggestions for Authors

The manuscript provides a comprehensive review of nano-aluminum preparation technologies, reactivity differences between nano- and micro-aluminum powders, their coating/modification strategies, and applications in explosives and propellants. The scope is valuable for the energetic materials community. However, several substantial issues limit its scientific rigor, presentation quality, and utility for readers. Addressing these would significantly improve the manuscript’s clarity, credibility, and impact. My specific comments are the following:

  1. The manuscript frequently repeats similar data and observations across subsections (especially within preparation methods and reactivity tests). For example, ignition delay times, heat flux density tests, and thermal analysis results are described multiple times for different authors without adequate synthesis. I suggest condense repetitive information by summarizing trends in tabulated or comparative form where possible.
  2. The review functions primarily as a literature compilation rather than an analytical synthesis. The authors rarely critically compare conflicting results, discuss limitations of methods, or propose explanations for observed trends (e.g., why some coating strategies outperform others). I suggest including a critical evaluation section for each major topic, identifying gaps, controversies, and directions for future research.
  3. While preparation methods are exhaustively listed, mechanistic explanations of particle formation, oxide layer behavior, or combustion enhancement from coatings (e.g., fluoropolymer, boron, GAP) are missing or superficial. Integrating of mechanistic discussions based on reaction thermodynamics, heat transfer, and particle morphology effects on combustion would be suggested.
  4. Continuing the previous comment, a recent paper presents a numerical model of the combustion mechanism of single micron-sized aluminum particles https://doi.org/10.1016/j.fpc.2024.05.006. I suggest inserting a short comment on the importance of this area of research and citing this paper.
  5. Although extensive ignition delay, combustion rate, and thermal data are presented, they are embedded in text, making it difficult for readers to compare values across studies. I suggest including comparative summary tables for ignition delay, activation energy, combustion enthalpy, etc.

Author Response

Response to Reviewer 2​​

​​    Comment 1:​The manuscript frequently repeats similar data and observations across subsections (especially within preparation methods and reactivity tests). For example, ignition delay times, heat flux density tests, and thermal analysis results are described multiple times for different authors without adequate synthesis. I suggest condense repetitive information by summarizing trends in tabulated or comparative form where possible.

​​    Response:​​ We agree with this judgment. When writing this manuscript, we are also prepared to write in way the reviewer’suggestion, and organize similar data into tabular form. However, when sorting out the data, we found that the raw materials and experimental methods used by different researchers are quite different. If we make our research conclusions based on this, the reliability cannot be guaranteed. Therefore, after our discussion, we organized this manuscript in the current form, presenting as much original data as possible, so that readers can judge and analyze the trend by themselves according to these data, and they may infer more accurate trends. Please understand.

Comment 2:​The review functions primarily as a literature compilation rather than an analytical synthesis. The authors rarely critically compare conflicting results, discuss limitations of methods, or propose explanations for observed trends (e.g., why some coating strategies outperform others). I suggest including a critical evaluation section for each major topic, identifying gaps, controversies, and directions for future research.

​​    Response:​​ Thank you very much for this question. Compared with the conventional micron aluminum powder, the research on nano aluminum powder is much less. Because the reactivity of nano aluminum powder is much higher than that of micron aluminum powder, it can spontaneously ignite even at room temperature. The effective aluminum content of nano-aluminum powder used by different researchers can be said to be very different, so the reliability of the conclusion is not completely convincing. But this research direction is very promising. It is for this reason that we didn't write a critical evaluation for each major topic, and pointed out the gaps, controversies and future research directions. The existing data can show that nano aluminum powder is more reactive than micron aluminum powder, which can improve the detonation performance of mixed explosives when used in explosives and improve the burning rate of propellants and the agglomeration of combustion products when used in propellants. However, there is still a lack of experimental data to qualitatively explain these facts.

​    Comment 3:​​While preparation methods are exhaustively listed, mechanistic explanations of particle formation, oxide layer behavior, or combustion enhancement from coatings (e.g., fluoropolymer, boron, GAP) are missing or superficial. Integrating of mechanistic discussions based on reaction thermodynamics, heat transfer, and particle morphology effects on combustion would be suggested.
​​    Response:​​ Thank you very much for this question raised by the reviewer, and we also appreciate this suggestion very much. Because the content of this manuscript is already very complicated, if we use a lot of space to discuss the reaction mechanism, reaction thermodynamics, reaction kinetics and other issues of each process, it will make the content of this manuscript extremely huge and messy. The original intention of writing this manuscript is to provide readers with as much original data as possible, so that they can get more valuable information so as to better design the next work plan, because few readers will collect and classify such a huge amount of data. In addition, as explained in the previous question, the state of nano-aluminum powder used by different researchers is very different, and it is almost impossible to draw a convincing conclusion if all kinds of mechanisms are forcibly explained. In any case, we sincerely thank the reviewer for this very valuable question.

​    Comment 4:​​Continuing the previous comment, a recent paper presents a numerical model of the combustion mechanism of single micron-sized aluminum particles https://doi.org/10.1016/j.fpc.2024.05.006. I suggest inserting a short comment on the importance of this area of research and citing this paper.

​​    Response:​​ This part has been added, and the suggested article has been quoted. Thank the reviewer for his suggestions.

​    Comment 5:​Although extensive ignition delay, combustion rate, and thermal data are presented, they are embedded in text, making it difficult for readers to compare values across studies. I suggest including comparative summary tables for ignition delay, activation energy, combustion enthalpy, etc.

​​    Response:​​ As explained above, when we first organized this manuscript, we planned to write it in the way suggested by this reviewer. However, when sorting out and analyzing the data, we found that the materials and experimental conditions used by different researchers are very different, and many researchers did not provide the specific state or effective aluminum content of nano-aluminum powder. In this context, if we compare and analyze the data lists of ignition delay time, activation energy and combustion enthalpy and put forward our views, we feel that the views based on this data are completely unconvincing. Therefore, this writing method was abandoned. This question raised by the reviewers is really very valuable, and we sincerely thank you. Please also understand.

Reviewer 3 Report

Comments and Suggestions for Authors

This paper reviews the preparation, reactivity, and applications of nano-aluminum powder in explosives and propellants. Various preparation methods including mechanical processes, evaporation-condensation, chemical reduction, and electrodeposition are discussed, each offering distinct advantages. Nano-aluminum exhibits significantly higher reactivity than micro-aluminum; however, it is prone to oxidation, which limits its efficiency. Therefore, protective coatings or surface modifications are necessary to enhance its stability and performance. The study emphasizes the need for further research into the preparation and storage of high-purity nano-aluminum powder.

The review article addresses an important topic and is well organized. However, before it can be considered for final acceptance and publication, the following minor revisions should be made:

  1. An abbreviation section should be added.

  2. A detailed table should be included to summarize the various applications of nano-aluminum in explosives and propellants. Additionally, the performance of nano-aluminum should be compared with that of other additives used in propulsion.

  3. A figure illustrating the difference in particle sizes of aluminum and its impact on the reactivity of propellants should be provided.

Author Response

Response to Reviewer 3​​

Comment 1:​ An abbreviation section should be added.

​​    Response:​​ Thanks to the reviewer's suggestion, the list of abbreviations has been added.

Comment 2:​A detailed table should be included to summarize the various applications of nano-aluminum in explosives and propellants. Additionally, the performance of nano-aluminum should be compared with that of other additives used in propulsion.

​​    Response:​​ We agree with this critique. ​A detailed table has been prepared to summarize the various applications of nano-aluminum in explosives and propellants.

    As for another problem, there are many kinds of additives used in propellant. Comparing the effects of nano-aluminum particles on propellant properties with those of other additives should be the subject of another paper. Please understand.

Comment 3: ​A figure illustrating the difference in particle sizes of aluminum and its impact on the reactivity of propellants should be provided.

​​    Response:​​ Thanks to the reviewer for this suggestion. First of all, the influence of aluminum particles with different particle sizes on propellant properties is not very closely related to the theme of this manuscript. Secondly, because the citation of pictures needs the authorization of the original journal, we have not had enough time to finish it. Please understand.

Round 2

Reviewer 1 Report

Comments and Suggestions for Authors

Nothing has changed in terms of literature sources. Moreover, one more work by authors from China has been added. There is no overview of works from the rest of the world.

Author Response

Nothing has changed in terms of literature sources. Moreover, one more work by authors from China has been added. There is no overview of works from the rest of the world.

Reply: 

Dear Editor and Reviewers,​​

    We sincerely thank you and the reviewer for your view on our manuscript titled "Preparation Technology, Reactivity and Applications in Explosives and Propellants of Nano-Aluminum: A Review" ( Manuscript ID: nanomaterials-3735887).

    Reviewer #1 maintains his/her opinion of a very parochial list of references. It is certainly true that the Chinese contribution to the topic is very significant. For this opinion, we agree. 

    Reviewer #1 request that the authors must provide proper recognition to the work done elsewhere. It is not such a difficult-to-comply-with request!

    For this opinion, we have searched all the important databases, including but not limited to SCI, EI, Elsevier, Wiley, Springer and CNKI. we have read all the manuscripts carefully and thoroughly to sort out this review.

     Please do not hesitate to contact us if further clarifications are needed.

​​    Thank you again for your time and guidance.​​

​​Sincerely,​​
​​                                                               Team of the authors

                                                                       2025-08-18

Reviewer 2 Report

Comments and Suggestions for Authors

The authors have performed a thorough revision of the original manuscript. All my critical comments have been addressed and properly responded to. As a result, I believe the revised manuscript has gained greater scientific rigor and clarity. Therefore, I can recommend it for publication in its present form.

Author Response

Reviewer #1 maintains his/her opinion of a very parochial list of references. It is certainly true that the Chinese contribution to the topic is very significant, but the authors must provide proper recognition to the work done elsewhere. It is not such a difficult-to-comply-with request!

Authors’ reply to Academic Editor

Dear Editor and Reviewers,​​

    We sincerely thank you and the reviewer for your view on our manuscript titled "Preparation Technology, Reactivity and Applications in Explosives and Propellants of Nano-Aluminum: A Review" ( Manuscript ID: nanomaterials-3735887). Reviewer #1 maintains his/her opinion of a very parochial list of references. It is certainly true that the Chinese contribution to the topic is very significant.

   For this opinion, we agree. 

  Reviewer #1 request that the authors must provide proper recognition to the work done elsewhere. It is not such a difficult-to-comply-with request!

  For this opinion, we have searched all the important databases, including but not limited to SCI, EI, Elsevier, Wiley, Springer and CNKI, and read all the manuscripts carefully to sort out this review. The paper is data-driven, hoping to get the understanding of the reviewer.

  Please do not hesitate to contact us if further clarifications are needed.

​​    Thank you again for your time and guidance.​​

​​Sincerely,​​
​​                                                               Team of the authors

                                                                        2025-09-02